# Weight Management for Students with Attention-Deficit Hyperactivity Disorder (ADHD): A Qualitative Study

**DOI:** 10.3390/healthcare10112225

**Published:** 2022-11-07

**Authors:** Ayelet Siman Tov, Inbal Halevi Hochwald, Riki Tesler, Gizell Green

**Affiliations:** 1Kibbutzim College of Education, Technology, and the Arts, Tel Aviv-Yafo 6250769, Israel; 2The Max Stern Yezreel Valley College, Yezreel Valley 1930600, Israel; 3Health Management Department, Ariel University, Ariel 4076405, Israel; 4Health Promotion & Wellbeing Research Center, Ariel University, Ariel 4076405, Israel; 5Nursing Department, Ariel University, Ariel 4076405, Israel

**Keywords:** obesity/overweight, ADHD, students, health behaviors, coping strategies, weight management

## Abstract

(1) Background: Individuals with Attention-Deficit Hyperactivity Disorder (ADHD) are more likely to respond with an ineffective coping behavior, combined with an increased risk of weight gain and unhealthy food consumption. The aim of the study was to examine coping strategies adopted by overweight adults with ADHD to promote healthy behaviors and weight-control management. (2) Methods: Descriptive qualitative research using semi-structured interviews analyzed through a thematic content-analysis approach. The study included 30 college students diagnosed with ADHD, with a BMI ≥ 25.5, who had lost ≥10 kg within at least one year and successfully maintained that weight for the past three years. (3) Results: The study yielded two main themes. The first is struggling with disappointments and negative feelings in the past, and the second is the reciprocity between weight management and coping skills strategies. The second theme includes three subthemes—cognitive strategies, behavioral strategies, and emotional strategies. (4) Conclusions: It is essential to understand the variety of coping strategies used by adults to cope with ADHD challenges that can potentially improve weight control management and healthy behaviors to design better, more accurate interventions, promoting the target population’s well-being and health.

## 1. Introduction

From children aged 6–12 years to adults with ADHD, there was an increasing trend in the prevalence of unhealthy weight (obesity: from 13.5% to 19.3%, overweight: from 18.8% to 31.2%) [1]. Meta-analyses confirmed that the association between ADHD and obesity was more substantial in adults, demonstrating that the risk of obesity increases in people with ADHD from childhood to adulthood [2,3].

Obesity is linked to significant illness at all ages and constitutes one of the risk factors for severe morbidity and mortality [4]. The association between obesity and attention-deficit hyperactivity disorder (ADHD) has been reported in research. People with ADHD have a higher BMI and are at a higher risk of gaining weight, a condition that rises with age [5].

ADHD is a neurodevelopmental disorder involving a lack of attentiveness, hyperactivity, and impulsivity, which are detrimental to one’s functioning at school and in social and employment settings [6]. Comparing adults with and without ADHD presents a similar picture regarding eating habits (i.e., amount of food, number of meals, daily calorie consumption). Still, people with ADHD prefer eating more fattening and unhealthy foods (e.g., biscuits, processed meat, processed foods) and less healthy foods (e.g., fruit, vegetables, whole grains) [7].

A strong link between ADHD and a less healthy lifestyle was found, as evidenced by the younger adults engaging in fewer healthy lifestyle behaviors than the control group of the same age. These findings held even after controlling for multiple confounding variables such as age, gender, IQ, ADHD medication, comorbid psychiatric conditions, and household income [2,8]. Healthy behaviors can influence one another, and adults with ADHD who exhibit one unhealthy behavior are more likely to exhibit other unhealthy behaviors [9,10,11].

People with ADHD have difficulty developing adequate social support and asking for help from others, which prevents them from coping effectively using methods based on planning and problem-solving strategies [12]. Coping is an expression of consistent and cognitive behaviors and efforts that constantly change and are acted upon by an individual to generate change and remedy any imbalance resulting from an unusual external or internal occurrence [13]. Lazarus (2001) discerns between two main coping styles. The first is the problem-focused coping style, relying on active cognitive and behavioral strategies to change or address the sources of stress. Versions of problem-focused coping are usually chosen when an individual believes that the situation or challenge can be changed. These problem-focused solutions promote one’s mental well-being and personal growth [14]. The second coping style is the emotion-focused coping style, aiming at controlling emotional reactions resulting from the encounter with stress but without changing reality or the stressful situation, such as positive thinking, avoidance, or denial. Emotion-based coping will usually be chosen when the individual believes the threatening or stressful environmental conditions cannot be changed or when the individual must first calm down to act and apply a problem-solving coping strategy [15].

Effective coping with stress causes and managing challenging situations requires a continuous adaptation of proactive cognitive and behavioral efforts, such as planning and problem-solving [16]. However, people with ADHD find it harder to change according to the situation [17]; have fewer coping resources for stressful, challenging situations; tend not to use effective strategies due to their inattentiveness and impulsivity [18]; or use ineffective coping styles (i.e., disengagement, denial, or avoidance) [19]. Poor planning is also connected with impulsive reactions, causing one to choose an ineffective coping style, such as avoiding planning [20].

Considering these findings, showing the poor adjustments and ineffective coping of people with ADHD, combined with an increased risk of weight gain and unhealthy food consumption, leads to this study’s primary aim—to examine coping strategies and their implications for promoting healthy behaviors and weight-control management adopted by overweight adults with ADHD.

## 2. Materials and Methods

### 2.1. Research Design

A descriptive qualitative phenomenological study was conducted using content analysis. This method was used for making replicable and valid inferences from the data.

### 2.2. Participants’ Recruitment

Inclusion criteria were adults with ADHD who were previously diagnosed as overweight [BMI ≥ 25.5] and managed to lose weight and maintain a normal BMI for the past three years. Adults who were not once overweight or who did not have ADHD problems were excluded.

The study included 30 college students diagnosed with ADHD, with a BMI ≥ 25.5 (7 were overweight, and two were obese (BMI ≥ 30), who had lost ≥10 kg within at least one year (10–40 kg with an average loss of 20 kg) (all of the information was self-reported by the participants and was documented by the interviewer) and have successfully managed by maintaining that weight for the past three years. Interviewees were mostly single Israeli Jewish women, with an age average of 28.6 (ranging from 24 to 46).

### 2.3. Recruitment and Procedure

A letter was sent to all preservice students in one academic institute. The researcher contacted students interested in scheduling an interview and explaining the study’s aims. The students were promised anonymity and assured that the data would be used solely for research purposes.

### 2.4. Data Analysis

All participants gave written informed consent after receiving oral and written information. The study was approved by the Kibbutzim College of Education, Technology, and the Arts ethical IRB (Institute Review Board).

Following this procedure, thirty open in-depth interviews were conducted by a single researcher, with the main research question dealing with the interviewees’ coping strategies and their implications for promoting healthy behaviors and weight-control management throughout their lives. In addition, we wanted to find out how the two coping strategies influenced each other and what was the reciprocity between them.

During the interview, the researcher asked in-depth probing questions [21]. In addition, each interviewee was asked for sociodemographic characteristics and data regarding weight and state of health. This methodology enabled an in-depth understanding of the research questions [22]. The interviews lasted approximately 90 min and were recorded and transcribed. Content analysis was performed by two independent reviewers, experts in healthy behaviors. The analysis involved reading each text separately, isolating parts of the text, classifying and grouping them into categories, and analyzing them. After identifying the content categories, content and statements were located and grouped to create key themes [23]. Finally, a lateral theme analysis was conducted, cross-referencing the content categories identified from the data to find themes and create umbrella themes [24].

## 3. Results

The interviewees addressed several topics related to their experiences and coping strategies in the context of ADHD, health, and eating behaviors in the past and following weight loss. One of the main themes that emerged in the interviews was struggling with disappointments and negative feelings in the past. This theme describes the negative emotions before weight loss, compensated through emotional eating. The second theme addressed the present use of coping strategies and reciprocity between weight management and learning skills strategies. This theme deals with the bi-directional effects of the strategies adopted by the interviewees for the benefit of both academic studies and weight loss. The subthemes are: (A) cognitive strategies, (B) behavioral strategies, and (C) emotional strategies (See Figure 1).

### 3.1. Struggling with Disappointments and Negative Feelings in the Past

The interviewees were asked about their past experiences coping with a healthy diet and weight loss. The majority described a struggle and frustration with weight loss and body image. It is interesting to observe the changes after losing weight and succeeding in maintaining it, as a significant event that divided their emotional experience into a ‘before’ and ‘after’ experience.

The interviewees described their life in the past as a negative, maladaptive period characterized by self-criticism and self-blaming. Eating was described as emotional eating and a solution to frustration, stress, and difficulties in their lives:


*“I used to cope with situations such as exams, which brought strong negative emotions. Food used to calm me down. Food was ‘my best friend’ and a source of comfort and reduction of fear, tension, and stress in moments of distress.”*


The interviewees used to fantasize about comforting nutrients (sugar and carbohydrates). Comforting food was present in their thoughts and provoked negative emotions and eventually emotional eating:


*“In the past, I tended to fantasize about and imagine high-fat and high-sugar foods, such as cakes and pastries, almost automatically, and in the end, I gave up and ate them.”*


One of the challenges described was the setback between times of success, self-worth, and achievement to times of failure, guilt, and shame when not meeting their expectations. Some related to themselves in a dichotomic way: success and joy versus failure and guilt. One of the interviewees described himself as a failure, which led to a recursive loop of continued deterioration and eating. The inner talk was characterized by guilt, failure, and incapacity:


*“Whenever I carried out the eating plan perfectly, I felt successful and valuable, and every time I ate something that wasn’t on the meal plan, I felt like a complete failure and self-blamed, like- you’re not worth anything, you’re damaged and incapable.”*


Although the interviewees demonstrated struggling with disappointments and negative feelings in the past, in the present, they emphasized strength by managing those by using the reciprocity of coping strategies, leading to the second main theme.

### 3.2. Reciprocity between Weight Management and Learning Skills Strategies

This theme deals with the mutual effects of the strategies adopted by the interviewees for the benefit of both their studies and weight loss. The descriptions revealed three broad sub-themes of coping strategies: (1) cognitive strategies, (2) behavioral strategies, and (3) emotional strategies.

The coping process and the learning strategies that the interviewees learned regarding health behavior were used for other areas in their lives, such as academic abilities, and vice versa; skills learned to improve their studies due to ADHD were utilized for weight loss.


**A.** 
**Cognitive Coping Strategies**



One of the changes in the interviewees’ ways of thinking after they lost weight and maintained the new weight over time was a change from dichotomous thinking, characterized by criticism and self-blame, to a regulated mindset, with an understanding of their abilities and limitations, and acceptance of the path that includes ups and downs.


*“Nowadays, I see coping as a continuous process that has ups and downs, and I try to see my coping as containing different shades of grey and in terms of progress.”*


In this process, they expressed self-compassion, curiosity, and a desire to learn about their abilities and how they will work better for them. This change involved using proactive language based on choices, emphasizing self-efficacy, growth, and hope. The shift in attention led to a broad general change in perception perceiving sources of strength as essential and focusing on them more than on the origins of difficulty:


*“In the past, I used to focus on my suffering and on what I didn’t have, and I gained weight because others hurt me. But now, there is the language of action, and efficacy, focusing on positivity, optimism, and hope. I’ve become more active, cooking, and preparing food in advance and exercising according to a schedule.”*



*“I noticed that nowadays, I focus on positive feelings and achievements I have attained. I am experiencing a similar change in the academic field. I focus less on the struggles I had experienced in the past due to my ADHD and focus on the strengths that have led me to success and achievements today.”*


The following quotes provide an additional example demonstrating a process of learning from experiences and reflective thinking regarding what helps her move forward and what holds her back. Unlike the past, this process involves less self-judgment and more self-acceptance and compassion:


*“I’ve matured, and today, I am open to observing the unhealthy behaviors I had in the past. In the past, if one day I overate, then I would fast the next day. Today, when I overeat, I try to put myself together and continue with the healthy and balanced eating plan as usual.”*


Most interviewees described a different kind of attentiveness to themselves as a way of identifying the physiological feelings of hunger and repletion, and the factors and risky situations that led them to overeat:


*“Being attentive to myself also helped me discern real feelings of hunger and repletion…. This inner discussion helps me to identify what causes me to overeat. For example, watching chocolate commercials and smelling pastries are risky for me.”*


This attentiveness has influenced their priorities and has made them put their healthy lifestyle and own needs at the center:


*“My weight change is due to a deep understanding of what is essential in life. For the first time in my life, I put myself and my health first and committed to taking actions that support this value.”*


Most interviewees described a sense of self-efficacy experienced in implementing healthy behaviors and weight control that influenced other areas of life:


*“I have proved to myself that I can finally make myself feel good and promote my health… I finally see that I can fill that void in me because of the failures I’ve experienced over the years.”*



*“The feeling of success in my process of weight-control and change of eating habits makes me feel good and encourages me to believe I can successfully cope with other challenges in my life.”*



**B.** 
**Behavioral Coping Strategies**



The interviewees described setting up the home environment to fit their health-behavior needs by increasing the amount and accessibility of healthy, tasty, and enjoyable foods:


*“The first stage of adopting healthy behaviors was changing the foods I brought into my home. I identified healthy foods that I find tasty and that give me pleasure and satisfaction.”*


The interviewees reported being careful to persevere with their new eating habits consistently. They compared their successes to past failures, not necessarily in areas of health behavior, but also from other areas such as success and perseverance in their studies:


*“My current journey shows abilities that I didn’t know I had, like being able to persevere. For example, if I didn’t finish my studies, I quit. But nowadays, for weight-maintenance and control, I find myself striving, coping, and succeeding.”*


Interviewees applied strategies such as scheduling nutritious meals (i.e., fruit and vegetables, water, little fat, sugar, and carbohydrates), eating at regular intervals, decreasing unhealthy foods, and documenting and monitoring eating habits in a daily journal. This habit was also applied to studying skills habits:


*“Using a journal to write my eating gives me immediate feedback to control consumption and plan the rest of the day accordingly.”*



*“Documenting my eating in the journal allows me to create an inner dialogue and take care of the situation in real time. I find myself, albeit partially, managing to control other areas that are related to studying.”*


Exercising was one of the main elements in developing the new health-promoting mindset they adopted during the coping process toward a healthier lifestyle:


*“Promoting health isn’t just weight-control; instead, it’s about promoting a healthy lifestyle that must include exercise.”*


Most interviewees reported using the planning strategies they had learned for eating in other areas of their life, such as studying and improved attentiveness:


*“I use learning management skills, like creating a list of necessary ingredients. Thus, I’ve experienced a meaningful improvement in my organizational skills. From organizing material for an exam to organizing documents connected to my household affairs. I can better distinguish between what is essential and what isn’t.”*


Some interviewees reported they had adopted a mindset of examining their eating behaviors through critical reflection, or like a researcher studying phenomena in the world, looking at the processes. They had developed the ability to control and correct their eating behavior while it was still happening, a power they also partially applied to other areas of life:


*“I ask myself questions, like, what happened? What led me to deviate from the original plan? What do I really need right now? Is it really a particular food or something else? If the answer is still that I need food, I ask myself what type of food it will be, so it will give me pleasure and repletion.”*



*“I feel that this skill of reflecting and asking questions about my eating helps me cope with other areas, such as interpersonal conflicts, relations with professors, friends, and my partner.”*



**C.** 
**Emotional Coping Strategies**



As described in the first theme, many interviewees described emotional difficulties in the past and emotional eating as one of the main coping mechanisms they used to deal with this struggle. Most interviewees mentioned they had to identify and use sources of relaxation and pleasure other than food to help them cope with stress, anxiety, and situations where the impulse to eat arises:


*“In the past, I ate whenever I felt anxious or stressed. Nowadays, when I think about the urge to eat, I turn to activities that please me, such as drawing mandalas. This activity helps me to be with myself, identify the cause of the distress or anxiety and calm down.”*



*“During the current change, I understood that there is no real danger; I was more scared of the discomfort of being stressed than of the actual occurrence itself.”*


The interviewees reported varied sources of learning the strategies, some formal, such as weight-loss groups, and others informal, such as family and friends. All the interviewees took part in some therapy process throughout life, whether for weight loss or learning strategies within the framework of their studies. They found those groups to be support groups, not only as a source of information and development:


*“In the support group, I received encouragement and compliments from the other participants. They strengthened my self-efficacy and my self-belief and made me feel very good about myself. When I feel good about myself, I manage to lose weight.”*


Some interviewees noted that looking at their past photos when they were overweight helped them feel happier and motivated them to continue the changes in their behavior:


*“Every few months, I look at pictures from my fat period and remember the hard feeling I experienced, and this way, I reinforce myself to continue and change my situation.”*


The interviewees noted that success and maintaining weight loss had improved them in many other areas of life—emotional, physical, academic, and social.


*“I feel good about myself, also because the experience of success adds to my motivation. I have more energy and physical strength to perform tasks in my life than before. I feel lighter, more awake, and vital, and I am more willing to perform tasks in other areas of life that I used to put off until the last minute.”*


## 4. Discussion

When compared to other age groups, young adults (18–35 years) have the highest rates of weight gain [1,2], as well as rising rates of cardiovascular disease, diabetes, and cancer risk. Obesity is expected to increase the public health burden among young adults, emphasizing the need for weight-related interventions [3]. Moreover, according to research from the past 15 years, one factor that increases the risk of obesity may be ADHD [5]. According to available sources, ADHD is a significant risk factor for obesity, which is especially visible in the adult population [4,5,6,7]. Many studies have dealt with people’s coping strategies regarding weight management and ADHD [3,4,5,6].

Compared to healthy people, the risk of obesity is 40% greater, and the risk of being overweight is 50% higher in those with ADHD. From a lack of dependence or reduced prevalence of obesity among young children with ADHD compared to their peers to a high incidence of obesity in adults, the association between ADHD and obesity is increasing with age [2].

This study explores and describes the experiences and challenges facing adults with ADHD regarding their health behavior and unpacks the coping strategies they use to achieve healthy behavior and lose weight.

This study offers a dual innovation in the then-and-now description of strategy coping for weight management, the dyadic implication of one field on the other, for better coping abilities and adherence to changes over time and with better results.

The first theme describes the past challenges, struggles, disappointments, and negative feelings shared by interviewees with ADHD and obesity. In the second theme, the interviewees describe changes in their state of mind, emotions, thoughts, and behavior due to the adoption of healthy behavior strategies as part of the weight-loss process or as part of adopting good learning strategies and skills during their academic studies. Coping strategies complement one another and render each other more effective. Coping is not seen as one action but rather as a complex system of activities that changes according to the situation [14].

One of the main themes described by most interviewees deals with struggling with disappointments and negative feelings in the past. The interviewees described maladaptive periods characterized by self-criticism and self-blaming. Eating was described as a positive, comforting experience and an emotional act, saving them from frustration, stress, and a hard time.

ADHD is associated with disordered eating, especially addictive-like eating behavior (i.e., binge eating, food addiction, and loss of control overeating). ADHD and addictive-like eating behavior are both associated with negative affectivity and emotion dysregulation, which we hypothesized to act as mediators of this relationship [23]. ADHD can significantly impact many aspects of adult life, from social and emotional well-being to professional development and financial security [24]. Depression, low self-esteem, and anxiety are correlated with ADHD [25] and were identified as the strongest predictors of poor quality of life for adults with ADHD [26]. Adults with ADHD are more likely to have somatic disorders [27], obesity, and sleep disorders [28].

Those challenges, together with ADHD and eating habits, led the interviewees, who were able to lose weight and maintain the new weight over time, to develop different coping strategies. Those strategies had specific implications for coping with difficulties stemming from ADHD and weight management.

In this study, adopting healthy behavior was found to have a further contribution to the participants’ executive functioning and emotional regulation. Executive functioning refers to conscious and controlled neurocognitive processes which support organized and goal-oriented activity, thinking, and behavior [29]. Individuals with ADHD display poorer performance in these functions [30]. This corresponds to this study’s findings since most interviewees reported that the strategies they had adopted to enhance healthy behaviors (i.e., food journal and home food environment) contributed to their functioning in the areas of planning, organizing, asking critical questions, self-control, and stopping and correcting a behavior while it was still happening. This non-judgmental approach replaced their former perspectives, which led to feelings of self-criticism and self-blaming. Difficulties in implementing these functions are typical of people with ADHD and may impact their daily functioning [31].

The second main theme—reciprocity between weight management and learning-skills strategies—yielded three effective coping strategies used to promote healthy behaviors and weight control: cognitive, behavioral, and emotional coping strategies.

The cognitive strategies identified in this study helped the interviewees to reassess and re-examine situations that aroused negative emotions. They changed their dichotomous thinking, characterized by criticism and self-blame, to a cognitive, flexible, balanced mindset, with an empathic approach to their abilities and limitations, accepting the path that includes ups and downs. This cognitive restructuring allows them to re-evaluate the situation, develop a new, more adaptable perspective of the threatening emotion, and create a mental distance from it.

The coping strategies described in this study correspond to the cognitive–behavioral approach described in the literature, which focuses on changing a person’s ways of thinking in order to bring about the desired emotional and functional change, as well as a change in the quality of life [32]. This approach is based on identifying maladaptive cognitions and thought patterns and then carefully and rationally examining their validity and accuracy. This process strengthened the interviewees’ ability to endure an internal adverse event and experience it as tolerable (distress tolerance). This may lead to more rational and appropriate perceptions. The cognitive-behavioral approach was found to be effective for weight control and weight preservation over time and in treating eating disorders [33], but also as having moderate benefits for treating adults with ADHD, as well as adults with ADHD’s secondary disturbances, such as depression, and anxiety [34].

Another cognitive coping strategy identified in this study was the interviewee’s non-judgmental attention to their thoughts, including identifying and paying attention to physiological hints of hunger and repletion. Turning one’s attention to bodily processes, allowing for flexibility, and not ignoring them can deliberately focus on healthy foods. This strategy is typical of mindfulness techniques, defined as one’s awareness of the present moment by turning one’s attention to one’s current feelings and thoughts, deliberately and without judgment, and without trying to evaluate or change them [35].

Regarding the behavioral coping strategies, the interviewees described many behavioral adjustments to cope with their wish to lose and maintain weight. All the interviewees mentioned that they consistently adjusted their home food environment (i.e., accessible healthy food and reducing stored unhealthy nutrition). Other researchers also found that building a personal eating environment and setting up strategic structures that support a healthy lifestyle contributed to hunger prevention during weight control and maintaining weight loss [36,37,38].

Food journal and physical activity were the main tools described in the interviews for monitoring food consumption and weight loss. The journal includes cognitive and behavioral components of planning and executing food consumption and monitoring feelings of hunger and repletion. Food journal is one of the most effective strategies for identifying stressors along with antecedents and triggers for weight control and mindful eating [39]. The contribution of physical activity to weight loss and a healthier lifestyle has been widely documented in the literature [40,41], demonstrating a positive, long-lasting influence on well-being, metabolic profile, weight loss, and improved physical fitness [42]. Physical activity has also been shown to improve cognitive functions, such as academic performance, for people with and without ADHD [43] and to reduce the risk of depression, anxiety, and low self-esteem [44].

Regarding emotion-focused coping strategies, most interviewees reported using alternative resources, such as getting busy with a creative activity (drawing, writing, and dancing), as an effective coping strategy providing them with a powerfully positive experience. The literature describes this as a ‘flow’-a form of intense enjoyment and satisfaction when doing something with total involvement, such as artistic, game-like, or professional activities [45]. The fierce concentration associated with reduced perception of irrelevant stimuli and improved task performance [46] is effective for eating disorders, including binge eating [47]. Using alternative forms of relaxation allowed the interviewees to experience interest and pleasure and have rewarding and balanced experiences without unpleasant emotions and/or unexpected impulses when directing their attention to eating, which increased the likelihood of accumulating other successful and enjoyable occasions in the future.

Another emotional strategy the interviewees described was being supported by family, friends, and support groups, which promoted joy, empathy, intimacy, encouragement, and a shared language addressing health promotion. This process strengthened the feelings of belonging, efficacy, and self-control. Social support is a barter system that provides material and physical-aid information about the environment and its resources, social and emotional ties, and a feeling of concern and caring [48]. The literature also emphasizes the contribution and effectivity of formal social support for weight loss, maintaining the loss over time [49], and a healthier lifestyle [50]. There are known advantages of group therapy for people with ADHD and for the support of coaching for students with ADHD symptoms and issues with executive functioning [51].

The interviewees also adopted a non-judgmental emotional regulation process by recognizing emotional events. Emotional regulation refers to the strategies through which people direct their emotional stimulation, control it, and change it, so they can function optimally by their goals, considering the social context [52]. Individuals with ADHD have greater difficulty regulating emotions, and non-regulated emotions are related to binge eating [53,54]. Indeed, the interviewees noted that they used the emotional regulation strategy to improve their learning skills and control their weight management.

The combination of the contribution of all three strategies: cognitive strategies (redirecting one’s focus and re-evaluating) [32], emotional processes (acceptance of reduced negative emotions—accepting inner events) [35], and behavioral strategies (distraction, exposure, and delaying a response) have a more significant effect on both the subjective experience and the behavioral and physiological response [32]. The study’s findings suggest that adopting stable coping strategies for healthy behaviors led to implementing those skills in other areas, such as academic skills and social and interpersonal relationships.

Those strategies revolve around one’s ability to be flexible. Psychological flexibility is considered essential to one’s mental health and has a multidimensional structure with cognitive, behavioral, and emotional aspects [36,38]. It is expressed through identifying and adapting to a different situation, changing behaviors and ways of thinking that damage one’s functioning, and being aware, open, and committed to appropriate behaviors that match one’s values [38,51]. Psychological flexibility is lower among individuals with ADHD [17]. The coping strategies adopted by the interviewees improved their psychological flexibility by recognizing their range of emotions, getting used to demands deriving from various situations, and changing psychological conditions (thoughts and feelings) or their behavioral repertoire as the context required for weight loss and for learning management.

The current study’s findings offer profound insight into the mechanisms contributing to actualizing psychological flexibility and behaviors promoting psychological well-being. These mechanisms mainly include changing the person’s acceptance and mindfulness, restructuring health as a top priority, and directing one’s attention to internal and external resources.

### 4.1. Strengths and Limitations

Our study had several strengths and limitations. The sample size was appropriate for a qualitative study but mainly included women. Furthermore, it cannot be assumed that the findings can be generalized or applied to other similar populations. However, the use of in-depth interactive interviews and the use of consensus meetings to verify the credibility of the data strengthen our results.

### 4.2. Implications for Future Research

Future research should include standard and medical recommendations for treating obesity and an interdisciplinary team of physicians, psychologists/health behavior experts, nutrition experts, and physiologists. References to nutrition expertise should be included in eating and health behavior recommendations, as well as food and nutrition discussion. Moreover, it is essential to understand the variety of coping strategies used by adults to cope with ADHD challenges that can potentially improve weight control management and healthy behaviors to design better, more accurate interventions, promoting the target population’s well-being and health.

Further research is needed in order to assess a larger sample of both genders, with diverse backgrounds, for a deeper understanding of the coping strategies and resources, with an emphasis on psychological flexibility and resilience.

## 5. Conclusions

Obesity is a common severe phenomenon causing significant morbidity and mortality, and ADHD is also a common phenomenon associated with obesity and impaired well-being. Both conditions provoke mental and physical coping that occasionally has successes, which are challenging to adhere to and preserve over time. This unique study describes the past experiences of people struggling with disappointments and their present resources for coping with these challenges through psychological flexibility for managing healthy behaviors and proper weight control and its contribution to fully functioning people in all areas.

## Figures and Tables

**Figure 1 healthcare-10-02225-f001:**
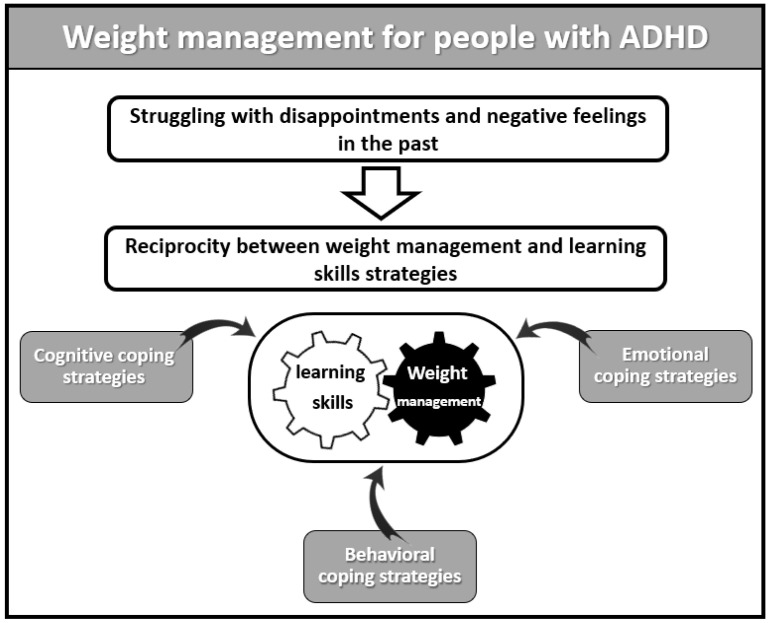
Outline of themes and subthemes.

## Data Availability

Not applicable.

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
