# Peer review of "Weight Management for Students with Attention-Deficit Hyperactivity Disorder (ADHD): A Qualitative Study"

_healthcare, 2022, doi:10.3390/healthcare10112225_

Round 1
Reviewer 1 Report
1. Do not use abbreviations in the title.: ADHD
2. The inclusion and exclusion criteria are not clear.
3. I suggest that the author could make a table and describe the experiences and challenges facing adults with ADHD regarding.
4. The discussion section is poor. The author has to explain the previous study and discuss that.
Author Response
Comments and Suggestions for Authors Number 1:
- Do not use abbreviations in the title.: ADHD
Thank you we added full name
- The inclusion and exclusion criteria are not clear.
Thank you we corrected the inclusion and exclusion criteria Lines: 74-78
- I suggest that the author could make a table and describe the experiences and challenges facing adults with ADHD regarding.
After more discussion with the authors, it was determined not to include a table in the article and to elaborate on the lines 41–46 that allude to this comment.
- The discussion section is poor. The author has to explain the previous study and discuss that.
Thank you; further references have been added to the discussion section.
LaRose JG, Leahey TM, Hill JO, Wing RR. Differences in motivations and weight loss behaviors in young adults and older adults in the National Weight Control Registry. Obesity. 2013 Mar;21(3):449-
Cortese S, Moreira-Maja CR, St Fleur D, Morcillo-Peñalver C, Rohde LA, Faraone SV. Association between ADHD and obesity: A systematic review and meta-analysis. Am. J. Psychiatry 2016; 173(1): 34–43. Doi: 101176/appi.ajp.2015.15020266.
Cregin D, Koltun R, Malik S, Umeozor D, Begdache L. The Adderall Epidemic: A Proposed Cyclic Relationship between ADHD Medication Use, Academic Performance, and Mental Distress.
Tate DF, Lytle L, Polzien K, Diamond M, Leonard KR, Jakicic JM, Johnson KC, Olson CM, Patrick K, Svetkey LP, Wing RR. Deconstructing weight management interventions for young adults: Looking inside the black box of the EARLY consortium trials. Obesity. 2019 Jul 1;27(7):1085-98.
Wing RR, Tate D, Espeland M, et al. Weight gain prevention in young adults: design of the study of novel approaches to weight gain prevention (SNAP) randomized controlled trial. BMC Public Health. 2013;13:300. [PubMed: 23556505]
Hanć T. ADHD as a risk factor for obesity. Current state of research. Psychiatr pol. 2018 Apr 30;52(2):309-22.
Altfas JR. Prevalence of attention deficit/hyperactivity disorder among adults in obesity treatment. BMC Psychiatry 2002; 2: 1–8. 9.
Fleming J, Levy L. Eating disorders in women with AD/HD. In: Quinn PO, Nadeau KG. Gender Issues and AD/HD: Research, Diagnosis, and Treatment. Silver Springs, MD: Advantage Books; 2002. p. 411–426.
Nigg JT, Johnstone JM, Musser ED, Long HG, Willoughby M, Shannon J. Attention-deficit/hyperactivity disorder (ADHD) and being overweight/obesity: new data and meta-analysis. Clin Psychol Rev 2016;43:67–79 [Available from: https://pubmed-ncbi-nlm-nihgov.ezproxy.unibo.it/26780581/] [Internet] [cited 2022 October 8].
Cortese S, Moreira-Maia CR, St Fleur D, Morcillo-Pe˜nalver C, Rohde LA, Faraone SV. Association between ADHD and obesity: a systematic review and meta-analysis [Internet]. Am J Psychiatr 2016;173:34–43 [Available from: https://ajp-psychiatryonlineorg.ezproxy.unibo.it/doi/abs/10.1176/appi.ajp.2015.15020266] [cited 2022 October 10].
Reviewer 2 Report
Article needs to be edited as follows;
1) Aim in abstract is not clearly stated
2)Sample in abstract is devoid of BMI and not reflective of true sample
3) conclusion is far stretch from the "themes" -
4)literature is weak - mentions "studies" but then lists only 1 ref. Lines 35-39 are unclear to me
5)Ref 3 is not an RCT- they reference it as if it is causal information
6) there is no clear inclusion and exclusion criteria - was this an after thought?
7)which institution gave IRB approval?
8) Method: "thirty open in-depth interviews were conducted by a single researcher with the main research question dealing with the interviewees' coping strategies and their implications for promoting healthy behaviours and weight-control management throughout their lives". Was any software used to identify key words and themes? What was the actual question?? Is not clearly stated. Theoretical framework is lacking
9)Lines 119-120 :Is this a conclusion that authors are making? seems to be a stretch in that this is a retrospective qualitative approach - not causal
10) seems to be one quote from one interviewee but then the authors state the finding as being from the group of thirty - was consensus reached from the 30 participants?
Author Response
Comments and Suggestions for Authors Number 2:
Thank you very much. We learned a lot from the Comments and Suggestions.
1) Aim in abstract is not clearly stated
Thank you; we have revised the study's aim.
2) Sample in abstract is devoid of BMI and not reflective of true sample
Thank you we correct the abstract lines: 14-15
3) conclusion is far stretch from the "themes"
Thank you; we have revised the study's conclusions lines:
4) literature is weak - mentions "studies" but then lists only 1 ref. Lines 35-39 are unclear to me
Thank you; we included more references lines: 35-39
LaRose JG, Leahey TM, Hill JO, Wing RR. Differences in motivations and weight loss behaviors in young adults and older adults in the National Weight Control Registry. Obesity. 2013 Mar;21(3):449-53.
Cortese S, Moreira-Maja CR, St Fleur D, Morcillo-Peñalver C, Rohde LA, Faraone SV. Association between ADHD and obesity: A systematic review and meta-analysis. Am. J. Psychiatry 2016; 173(1): 34–43. Doi: 101176/appi.ajp.2015.15020266.
Cregin D, Koltun R, Malik S, Umeozor D, Begdache L. The Adderall Epidemic: A Proposed Cyclic Relationship between ADHD Medication Use, Academic Performance, and Mental Distress.
Tate DF, Lytle L, Polzien K, Diamond M, Leonard KR, Jakicic JM, Johnson KC, Olson CM, Patrick K, Svetkey LP, Wing RR. Deconstructing weight management interventions for young adults: Looking inside the black box of the EARLY consortium trials. Obesity. 2019 Jul 1;27(7):1085-98.
Wing RR, Tate D, Espeland M, et al. Weight gain prevention in young adults: design of the study of novel approaches to weight gain prevention (SNAP) randomized controlled trial. BMC Public Health. 2013;13:300. [PubMed: 23556505]
Hanć T. ADHD as a risk factor for obesity. Current state of research. Psychiatr pol. 2018 Apr 30;52(2):309-22.
Altfas JR. Prevalence of attention deficit/hyperactivity disorder among adults in obesity treatment. BMC Psychiatry 2002; 2: 1–8. 9.
Fleming J, Levy L. Eating disorders in women with AD/HD. In: Quinn PO, Nadeau KG. Gender Issues and AD/HD: Research, Diagnosis, and Treatment. Silver Springs, MD: Advantage Books; 2002. p. 411–426.
Nigg JT, Johnstone JM, Musser ED, Long HG, Willoughby M, Shannon J. Attention-deficit/hyperactivity disorder (ADHD) and being overweight/obesity: new data and meta-analysis. Clin Psychol Rev 2016;43:67–79 [Available from: https://pubmed-ncbi-nlm-nihgov.ezproxy.unibo.it/26780581/] [Internet] [cited 2022 October 8].
Cortese S, Moreira-Maia CR, St Fleur D, Morcillo-Pe˜nalver C, Rohde LA, Faraone SV. Association between ADHD and obesity: a systematic review and meta-analysis [Internet]. Am J Psychiatr 2016;173:34–43 [Available from: https://ajp-psychiatryonlineorg.ezproxy.unibo.it/doi/abs/10.1176/appi.ajp.2015.15020266] [cited 2022 October 10].
5) Ref 3 is not an RCT- they reference it as if it is causal information
Thank you; we have corrected it.
6) there is no clear inclusion and exclusion criteria - was this an after thought?
Thank you we corrected the inclusion and exclusion criteria Lines: 74-78
7)which institution gave IRB approval?
The name of the institution is Kibbutzim College of Education, Technology, and the Arts, we added in line: 103-105
8) Method: "thirty open in-depth interviews were conducted by a single researcher with the main research question dealing with the interviewees' coping strategies and their implications for promoting healthy behaviors and weight-control management throughout their lives".
Was any software used to identify keywords and themes?
No dedicated software was used to analyze the content.
What was the actual question?? Is not clearly stated. Theoretical framework is lacking.
If and what coping strategies related to promoting healthy behaviors and weight-control management and their implications were adopted? Added, line 108-109 .
The Theoretical framework for the research question is derived from the research literature in the introduction section.
9) Lines 119-120:
Is this a conclusion that authors are making? seems to be a stretch in that this is a retrospective qualitative approach - not causal
Thank you very much for your comment, paragraph has been corrected
Finally, a lateral theme analysis was conducted, cross-referencing the content categories identified from the data to find themes and create umbrella themes [15]. Line 118-119.
10) seems to be one quote from one interviewee but then the authors state the finding as being from the group of thirty - was consensus reached from the 30 participants?
Thank you for your attention, there is a consensus. The first theme " Struggling with Disappointments and Negative Feelings in the Past" it is a relatively small theme than the second theme "Reciprocity between Weight Management and Learning Skills Strategies", therefore there is a total of three quotes for the theme. Each one is a representative quote that emphasizes, describes, and gives support to the developing theme. Regarding the second theme, there is more than one quote.
Reviewer 3 Report
This is a very interesting qualitative study that describes coping strategies related to eating behaviors and weight management in ADHD adults.
1. The authors note in Lines 29-31 that " The association between obesity and attention 29 deficit hyperactivity (ADHD) has been extensively reported in the literature. People with 30 ADHD have a higher BMI, and they are at a higher risk of gaining weight, a condition 31 that rises with age [3]."
However- only 1 article is cited, which does not represent an extensive reporting of this association. Please either provide more references that clearly indicate this established association between obesity and ADHD, or rephrase this to state this relationship is still being explored.
2. Please expand on your methods. The authors note that they recruited students who had a diagnosis of ADHD and who had " previously undergone one or two kinds of treatment at some point: psychotherapy and/or remedial teaching with provided by learning strategies experts" (Lines 75-80) Please provide more specific/detailed information regarding: how did the authors know the participants had been diagnosed with ADHD? How do the authors know that the participants received the treatment indicated and who were the 'experts' that provided this treatment? Did the authors have access to medical records? health/student records? If so- what kind review was done of these records?
3. Implications for future research should reference standard and medical recommendations for treatment of obesity- which includes an interdisciplinary team of physicians, psychology/health behavior, nutrition experts, physiology experts. Eating and health behavior recommendations and discussion related to food and nutrition should include references to nutrition expertise as well.
Author Response
Comments and Suggestions for Authors number 3:
This is a very interesting qualitative study that describes coping strategies related to eating behaviors and weight management in ADHD adults.
Thank you very much. We learned a lot from the Comments and Suggestions.
- The authors note in Lines 29-31 that " The association between obesity and attention 29 deficit hyperactivity (ADHD) has been extensively reported in the literature. People with 30 ADHD have a higher BMI, and they are at a higher risk of gaining weight, a condition 31 that rises with age [3]." However- only 1 article is cited, which does not represent an extensive reporting of this association. Please either provide more references that clearly indicate this established association between obesity and ADHD, or rephrase this to state this relationship is still being explored.
Thank you very much, it has been corrected lines 34-37
- Please expand on your methods. The authors note that they recruited students who had a diagnosis of ADHD and who had " previously undergone one or two kinds of treatment at some point: psychotherapy and/or remedial teaching with provided by learning strategies experts" (Lines 75-80) Please provide more specific/detailed information regarding: how did the authors know the participants had been diagnosed with ADHD? How do the authors know that the participants received the treatment indicated and who were the 'experts' that provided this treatment? Did the authors have access to medical records? health/student records? If so- what kind review was done of these records?
The authors did not have access to records. Corrected accordingly lines 88-92.
- Implications for future research should reference standard and medical recommendations for treatment of obesity- which includes an interdisciplinary team of physicians, psychology/health behavior, nutrition experts, physiology experts. Eating and health behavior recommendations and discussion related to food and nutrition should include references to nutrition expertise as well.
Thank you very much. Based on your recommendation, we have added the paragraph dealing with future recommendations
Round 2
Reviewer 3 Report
Thank you for the opportunity to review this revised article. This is an interesting and relevant topic. The authors addressed most previous comments, which has improved the quality and flow of the paper. Please see further recommendations below:
1. Would remove reference #3 or appropriately cite (Cregin D et al). This reference is not properly cited/has an incomplete format in the reference list. Based on review of the paper- this doesn't seem an appropriate reference related to obesity and ADHD, but it's more related to college students/mental health/inappropriate use of ADHD medications.
2. Thank you for clarifying recruitment methods. It sounds like 'pre-screening' for eligibility was done via a survey where participants self-reported having ADHD diagnosis. Please also clarify and include in your methods if participants' BMI and previous weight loss with ability to maintain that weight loss was asked in this survey- so that this information was self-reported as well? Please clarify for the reader how the researchers obtained information about participants' BMI, previous weight loss, and weight loss maintenance.
3. Please proof-read for all errors (grammatical, spelling, proper tense, capitalization etc.) For example: Line 33 "Meta-analysis- "M" should be capitalized as the beginning of the sentence. Line 92 seems like it should be: ..."managed to lose weight" (vs. managed to lost weight)
4. Recommend rewording the sentence from lines 112-113. As it currently reads, it is unclear what the authors are trying to state. "If and what coping strategies related to promoting healthy behaviors and weight-control management and their implications were adopted"
5. Please review reference list and please be sure all references are cited properly, and formatted according to journal guidelines.
Author Response
Comments and Suggestions for Authors
Thank you for the opportunity to review this revised article. This is an interesting and relevant topic. The authors addressed most previous comments, which has improved the quality and flow of the paper. Please see further recommendations below:
- Would remove reference #3 or appropriately cite (Cregin D et al). This reference is not properly cited/has an incomplete format in the reference list. Based on review of the paper- this doesn't seem an appropriate reference related to obesity and ADHD, but it's more related to college students/mental health/inappropriate use of ADHD medications.
The source has been removed from the article. There are enough scholars for validation of the information
- Thank you for clarifying recruitment methods. It sounds like 'pre-screening' for eligibility was done via a survey where participants self-reported having ADHD diagnosis. Please also clarify and include in your methods if participants' BMI and previous weight loss with ability to maintain that weight loss was asked in this survey- so that this information was self-reported as well? Please clarify for the reader how the researchers obtained information about participants' BMI, previous weight loss, and weight loss maintenance.
Changes were made-
The study included 30 college students diagnosed with ADHD, with a BMI ≥25.5, (7 were overweight and 23 were obese (BMI≥ 30), who had lost ≥10 kg within at least one year (10-40 kg with an average loss of 20 kg) (all of the information was self-reported by the participants and was documented by the interviewer) and have successfully managed to maintain that weight for the past three years. Interviewees were mostly single Israeli, Jewish women, with an age average of 28.6 (ranging from 24 to 46).
- Please proof-read for all errors (grammatical, spelling, proper tense, capitalization etc.) For example: Line 33 "Meta-analysis- "M" should be capitalized as the beginning of the sentence. Line 92 seems like it should be: ..."managed to lose weight" (vs. managed to lost weight)
Thank you. We thoroughly reviewed the article and made the changes.
- Recommend rewording the sentence from lines 112-113. As it currently reads, it is unclear what the authors are trying to state. "If and what coping strategies related to promoting healthy behaviors and weight-control management and their implications were adopted"
Thank you for your comment. The sentence was rephrased. The text now –
In addition, we wanted to find out how the two coping strategies influenced each other and what the reciprocity was between them.
- Please review reference list and please be sure all references are cited properly, and formatted according to journal guidelines.
All the references were formatted according to the journal's guidlines
